

# Measurement error using a SeeMaLab structured light 3D scanner against a Microscribe 3D digitizer

Dolores Messer[1], Michelle S. Svendsen[2], Anders Galatius[3], Morten T. Olsen[2], Vedrana A. Dahl[1], Knut Conradsen[1] and Anders B. Dahl[1]

[1] Department of Applied Mathematics and Computer Science, Technical University of Denmark, Kongens Lyngby, Denmark
[2] Globe Institute, University of Copenhagen, Copenhagen, Denmark
[3] Department of Bioscience, Aarhus University, Roskilde, Denmark

Corresponding author
Dolores Messer, dolmes@dtu.dk

## ABSTRACT

**Background**. Geometric morphometrics is a powerful approach to capture and quantify morphological shape variation. Both 3D digitizer arms and structured light surface scanners are portable, easy to use, and relatively cheap, which makes these two capturing devices obvious choices for geometric morphometrics. While digitizer arms have been the "gold standard", benefits of having full 3D models are manifold. We assessed the measurement error and investigate bias associated with the use of an open-source, high-resolution structured light scanner called SeeMaLab against the popular Microscribe 3D digitizer arm.

**Methodology**. The analyses were based on 22 grey seal (*Halichoerus grypus*) skulls. 31 fixed anatomical landmarks were annotated both directly using a Microscribe 3D digitizer and on reconstructed 3D digital models created from structured light surface scans. Each skull was scanned twice. Two operators annotated the landmarks, each twice on all the skulls and 3D models, allowing for the investigation of multiple sources of measurement error. We performed multiple Procrustes ANOVAs to compare the two devices in terms of within- and between-operator error, to quantify the measurement error induced by device, to compare between-device error with other sources of variation, and to assess the level of scanning-related error. We investigated the presence of general shape bias due to device and operator.

**Results**. Similar precision was obtained with both devices. If landmarks that were identified as less clearly defined and thus harder to place were omitted, the scanner pipeline would achieve higher precision than the digitizer. Between-operator error was biased and seemed to be smaller when using the scanner pipeline. There were systematic differences between devices, which was mainly driven by landmarks less clearly defined. The factors *device*, *operator* and *landmark replica* were all statistically significant and of similar size, but were minor sources of total shape variation, compared to the biological variation among grey seal skulls. The scanning-related error was small compared to all other error sources.

**Conclusions**. As the scanner showed precision similar to the digitizer, a scanner should be used if the advantages of obtaining detailed 3D models of a specimen are desired. To obtain high precision, a pre-study should be conducted to identify difficult landmarks. Due to the observed bias, data from different devices and/or operators should not be combined when the expected biological variation is small, without

testing the landmarks for repeatability across platforms and operators. For any study necessitating the combination of landmark measurements from different operators, the scanner pipeline will be better suited. The small scanning-related error indicates that by following the same scanning protocol, different operators can be involved in the scanning process without introducing significant error.

# INTRODUCTION

Geometric morphometrics (*Bookstein, 1998*; *Rohlf & Marcus, 1993*; *O'Higgins, 2000*; *Adams, Rohlf & Slice, 2004*; *Slice, 2005*; *Mitteroecker & Gunz, 2009*) has proven a valuable tool in ecological and evolutionary studies of e.g., phenotypic development (*Polly, 2008*) and population structure (e.g., *Galatius, Kinze & Teilmann, 2012*). Within the field of geometric morphometrics, there has been great interest in the development and availability of equipment for collecting 3D data. It is an acknowledged and widely used practice to obtain landmarks directly from a physical specimen through a three-dimensional (3D) digitizer arm—most commonly a Microscribe digitizer. In recent years, however, collecting landmarks on a 3D digital model, obtained by 3D scanning the physical specimen, has become a viable and widely accepted alternative. Such a pipeline offers many important advantages, namely the creation of a permanent digital model of a specimen, which can be stored, easily shared with other researchers, and used for further analyses or adjustments of the measurement protocol. Furthermore, the full digital 3D model offers the possibility to analyse shape using more advanced statistical methods compared to the analysis of the collected landmarks using a digitizer.

3D surface scanners create high-density point clouds of the scanned object's surface. They are portable, highly accurate and non-destructive, and are thus an obvious choice for 3D digitizing physical specimens, such as skulls and bones in museum collections. To confidently replace digitizer arms with 3D surface scanners for geometric morphometrics, it is important to assess the measurement error introduced by using 3D surface scanners and to compare the two devices.

Landmarks can be classified into three distinct types based on their anatomical locations and biological significance (*Bookstein, 1991*). Previous studies have compared the precision of Type I, II and III landmark measurements obtained with a digitizer and from surface scans of human crania. One study concluded that landmark measurements obtained with the two devices are comparable for clearly observed landmarks (*Algee-Hewitt & Wheat, 2016*), whereas another study revealed that landmark coordinates obtained with the digitizer are more precise on average, although the precision of the two devices varies considerably depending on the landmark type (*Sholts et al., 2011*). However, none of these studies investigated or quantified different sources of measurement error.

In 3D geometric morphometric studies, measurement error can be introduced at any phase, by several sources (*Fruciano, 2016*). There is measurement error from replicability of landmark placing (within-operator), and due to the involvement of multiple operators (between-operator). Technological sources of measurement error include the use of different devices (between-device), but also cover 3D model creation steps, such as scanner resolution and algorithms used for fitting and smoothing. Presentation error involves differences in the way specimens are positioned during data acquisition, and it includes measurement error due to optical distortion.

The only study which has addressed measurement error in geometric morphometric data collection involving both a digitizer and a surface scanner that we are aware of is *Robinson & Terhune (2017)*. These authors suggested that measuring landmarks based on 3D scans slightly increases repeatability and landmark precision compared to measurements with a digitizer, and that between-operator error is greater than between-device error, while within-operator error is similar among devices. However, landmark coordinate measurement is subject to a compounding error when using a digitizer, as an operator and the digitizer interact during the data collection. When comparing digitizers to surface scanners, it is therefore important to introduce the same type of error, which can be achieved by letting each operator separately scan the specimens, then letting them place landmarks on the reconstructed 3D models of their own scans.

In addition to surface scanners, there are other important 3D capture modalities that differ in their properties and come with specific advantages and disadvantages. Highly accurate micro-computed tomography (µCT) scanners generate a 3D volumetric model, and have been seen as the "gold standard" of 3D shape capture technology (e.g., *Marcy et al., 2018*). µCT scanners capture more information, such as internal structure, however, they are expensive and not portable, and the scanning process is usually time-intensive. Typically, µCT scanning will give a volumetric image where the number of voxels is proportional to the size of the detector. Thus, in order to image a large object in high resolution, multiple scans must be acquired and stitched together. The resolution is limited by the spot size of the X-ray (or neutron) source and by the physical setup of the imaging system. Moreover, µCT scanning might potentially lead to DNA damage (*Bertrand et al., 2015*; *Grieshaber et al., 2008*). *Marcy et al. (2018)* found that using surface scans generates more random within-operator error compared to µCT scans, however not accounting for multiple operator error, whereas *Shearer et al. (2017)* and *Giacomini et al. (2019)* found consistent operator error rates for surface and µCT scans.

Another inexpensive, portable and increasingly popular 3D capture modality is photogrammetry, which is based on aligning multiple 2D photographs into a 3D model. Previous studies found that photogrammetry can be seen as a reliable alternative to surface scanners, both in case of large skulls (*Katz & Friess, 2014*; *Evin et al., 2016*), and for smaller skulls, where the use of photogrammetry has only been assessed at interspecific level (*Giacomini et al., 2019*). However, photogrammetry involves time-consuming image processing compared to laser surface scanners (*Katz & Friess, 2014*), and it involves a higher risk of inconsistency or human error during photography (*Marcy et al., 2018*). *Giacomini et al. (2019)* report a lack of detail achieved for teeth reconstruction based on photogrammetry
as well as difficulties in reproducing thin structures, however, these issues can also occur when using surface scanners.

The presence of high random measurement error can obscure the biological signal under investigation through an increase in the amount of variance (*Arnqvist & Mårtensson, 1998*; *Fruciano, 2016*). As a consequence, measurement error is more serious when the biological variation of interest is relatively low. At interspecific level, most studies combining landmark data collected using different devices found that measurement error was sufficiently small compared to the biological variation (*Robinson & Terhune, 2017*; *Marcy et al., 2018*; *Giacomini et al., 2019*), whereas one study identifies a between-operator error similar to the level of biological variation (*Shearer et al., 2017*). At intraspecific level, *Robinson & Terhune (2017)* found that the biological variation is obscured by all considered types of measurement error. However, the exclusion of difficult to place landmarks can substantially reduce the amount of measurement error to low levels compared to intraspecific variation (*Fruciano et al., 2017*).

In addition to random measurement error, systematic error can be present in geometric morphometric datasets, changing the mean and introducing a bias to the data. As a result, differences induced by systematic error might be interpreted as differences between groups in the analysis (*Fruciano, 2016*). A few studies confirmed the presence of small amounts of bias considering different 3D capture modalities (*Fruciano et al., 2017*; *Marcy et al., 2018*), and 2D landmark data (*Fruciano et al., 2019*). However, none of the studies comparing a digitizer with 3D capture modalities for morphometric data collection have examined the systematic error introduced by device and operator.

The purpose of our study is to examine whether 3D surface scanners confidently can replace digitizers for geometric morphometrics. To this end, we quantified the levels of measurement error introduced by analysing 22 grey seal skulls using a Microscribe digitizer, and a 3D structured light scanning setup (SeeMaLab (*Eiriksson et al., 2016*)) for landmark measurement. Specifically, we assessed whether specimen shape can be sufficiently captured by the SeeMaLab scanner compared to a Microscribe digitizer, and we compared the level of measurement error to the biological variation at intraspecific level. In contrast to previous studies, we accounted for the compounding error of observer and machine, which arises when using a digitizer, by making different scans and thus 3D models of every specimen for the two operators. We further investigated whether there are systematic differences between the Microscribe digitizer and the SeeMaLab scanner, and between operators. The SeeMaLab scanner is assembled from low-cost, off-the-shelf parts, and it is an open source system (Technical University of Denmark, Kongens Lyngby, Denmark; https://eco3d.compute.dtu.dk/pages/3d_scanner; *Eiriksson et al., 2016*). In order to assess the level of error introduced by our scanning setup relative to other sources of measurement error, we additionally analysed an extended scanner dataset, where both operators were collecting landmark measurements on both reconstructed 3D models.

## MATERIALS AND METHODS

### Samples

The performance of the two devices was compared by capturing the shape variation of 22 grey seal skulls. Of these, seventeen skulls were held by the Natural History Museum of Denmark and originated from the Baltic Sea population, whereas five skulls originated from the western North Atlantic population and were held by the University of Helsinki in Finland (Table A1). All the selected specimens were intact, and did not have any abnormalities in size, shape or colour variation. We did not include mandibles.

### Generation of 3D models

The skulls were 3D scanned using the SeeMaLab scanner (*Eiriksson et al., 2016*). The scanner includes a rotation stage which allows for automatic rotation of the positioned skull and scanning from a set of predefined directions covering the full 360°. The SeeMaLab scanner produces a dense set of 3D points at each scan direction. Combining these sets results in a set of 3D points that is fully covering the skull in the given position. To capture the full shape, and to cover the landmarks clearly and as detailed as possible, the skulls were scanned in four clearly defined positions (Fig. 3 in Article S1), which were determined in a pre-study. Moreover, since individual skulls differ in colour, brightness and size, exposure time and the distance between cameras and rotation stage had to be adjusted for each skull to obtain a scan of good quality. Typically, exposure time was changed for each specimen, whereas the distance between cameras and rotation stage only was adjusted for larger size differences. As with any repeated measurement, we expected slight differences between scans of the same specimen due to differences in the scanning set-up. Those include for example variations in calibration or illuminations, and possibly also diverging choices between operators on exposure time and distance between cameras and rotation stage.

In comparison to commercial surface scanners, the 3D models were not automatically reconstructed, which allowed us to optimize the quality of the final 3D model by pre-processing the point sets and choosing the reconstruction algorithm. To avoid misalignment between scanning positions on the few skulls with loose teeth, the point measurements on loose teeth were removed manually from the point clouds from three positions using the open-source system MeshLab version 2016 (Italian National Research Council, Pisa, Italy; https://www.meshlab.net/; *Cignoni et al., 2008*). We further observed that the teeth sometimes were out of focus when scanning, which resulted in a point cloud with wider teeth. In such cases, the teeth were also removed manually from the resulting point cloud of a given position using MeshLab. On the basis of geometric features, the point clouds from the four positions of a given skull and scan were then globally aligned using the Open3D library (*Zhou, Park & Koltun, 2018*), followed by non-rigid alignment as suggested by *Gawrilowicz & Bærentzen (2019)*. The resulting point cloud was manually cleaned around the teeth in MeshLab to coarsely remove points resulting from reflection. The final 3D model was reconstructed on the basis of Poisson surface reconstruction (*Kazhdan, Bolitho & Hoppe, 2006*; *Kazhdan & Hoppe, 2013*) using the Adaptive Multigrid Solvers software version 12.00 by Kazhdan (Johns Hopkins University, Baltimore, MA, USA; http://www.cs.jhu.edu/~misha/Code/PoissonRecon/Version12.00). In order to assess

error introduced during 3D scanning, each skull was scanned by two different operators. All 3D models were generated by the same operator.

## Landmark data collection

For each of the 22 skulls, Cartesian coordinates of 31 fixed anatomical landmarks of Type I and II (Fig. 1; Table A2) were recorded by two operators. The choice of landmarks is inspired by previous morphometric studies of ringed seal (*Pusa hispida*) and Antarctic fur seal (*Arctocephalus gazella*) skulls (*Amano, Hayano & Miyazaki, 2002*; *Daneri et al., 2005*). Each of the two operators applied two different measurement methods: Direct measurement on the physical skulls using a Microscribe 3D digitizer (Immersion Corporation, San Jose, CA, USA; https://www.immersion.com; Fig. 1 in Article S1); and placement of the landmarks on the reconstructed 3D digital models of their own scan using the Stratovan Checkpoint software version 2018.08.07 (Stratovan Corporation, Davis, CA, USA; https://www.stratovan.com/products/checkpoint). As none of the landmarks were considered difficult to see or reach during data collection, landmarks were placed from a single perspective with the Microscribe digitizer. When using the Microscribe digitizer, the errors caused by the device itself and the operator are combined. Thus, in order to obtain comparable datasets from the Microscribe digitizer and the scanner pipeline, we made each operator digitize their own scans, combining any associated errors for the same observer in the scanner pipeline. Moreover, in order to assess within-operator error, each measurement session was completed twice by both operators. We considered two repeated measurements to be sufficient for estimation of raw, direct measurement errors. When using the Microscribe digitizer, landmark measurement on the same skull was repeated after a break of about twenty minutes, whereas in case of the scanner pipeline, landmark annotation was only repeated after having annotated at least two other specimens. The datasets obtained by operator A using a Microscribe digitizer were collected about one year earlier than all remaining datasets. This resulted in eight datasets, each containing landmark configurations from all 22 skulls, giving a total sample of 176 configurations of landmarks. We call the entirety of these datasets *Device Comparison Dataset*. Figure 2 gives an overview over the data acquisition process.

Finally, each operator was also annotating the reconstructed 3D models based on the scan from the other operator twice. Thus, for all specimens, we collected landmark measurements on the 3D models based on an operator's own scans, and on the 3D models based on the scans from the other operator. We call the entirety of these scanner datasets *Extended Scanner Dataset* (Fig. 3). In contrast to the data collection using the Microscribe digitizer, which is done directly on a physical skull, there are two steps involved when using the scanner pipeline: reconstruction of a 3D model, and placing landmarks on this digital model. Using the *Extended Scanner Dataset* allows us to disentangle the scanning-related error from the observer measurement error, and assess the level of error introduced by our scanning setup. A detailed data collection protocol is included in Article S1, allowing replication of our measurements.

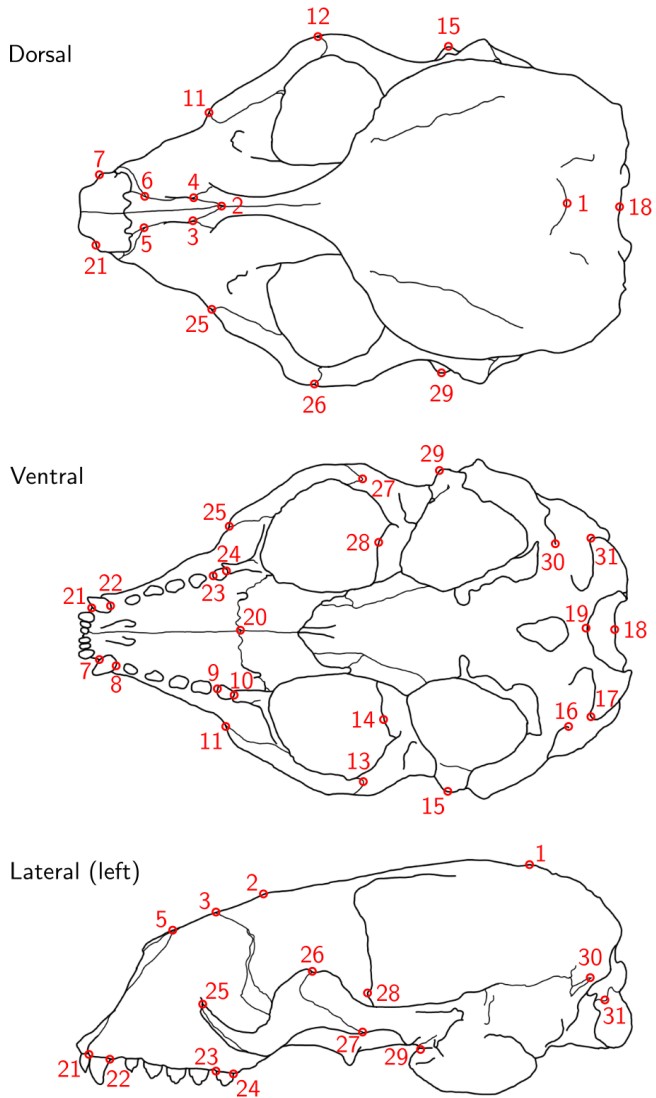

**Figure 1 Landmark definition.** The 31 anatomical landmarks measured on the grey seal skulls in this study. The choice of landmarks is inspired by previous morphometric studies of ringed and Antarctic fur seal skulls (*Amano, Hayano & Miyazaki, 2002*; *Daneri et al., 2005*). Based on original illustration of a ringed seal by *Amano, Hayano & Miyazaki (2002)*.

## Data analysis

For the data analysis, we chose 28 landmarks, as landmarks 16, 19 and 30 were non-reproducible here and therefore had to be omitted. Originally, the bilateral landmark pair 16/30 was defined on ringed seal skulls (*Amano, Hayano & Miyazaki, 2002*). However, on grey seal skulls, these landmarks turned out to not be as clearly defined, as the suture above the lateral apex of the condyle was often split into two sutures and thus did not have an unequivoval posteriordorsal apex. We did not notice before having collected all data that landmark 19 was non-reproducible in the case of using a Microscribe digitizer, which was caused by our measurement setup (Article S1, Fig. 1), where landmark 19 was slightly
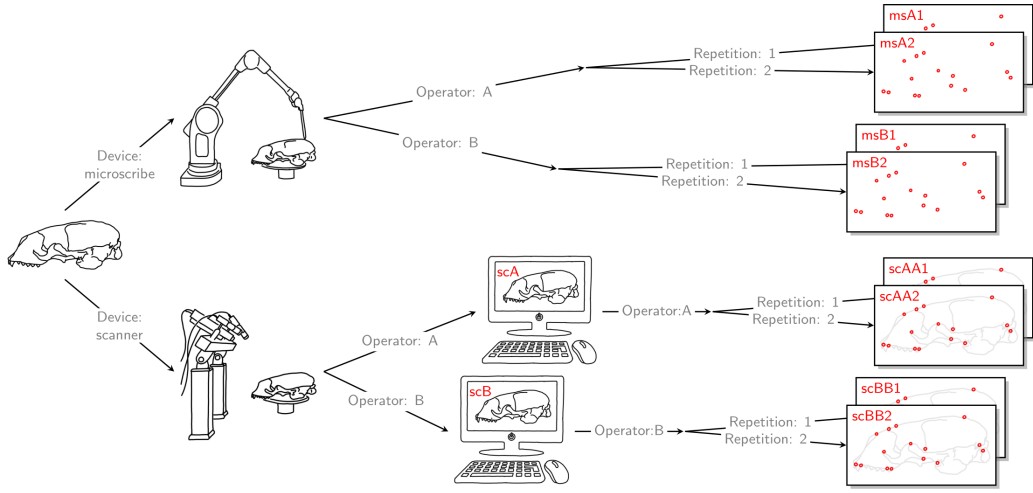

**Figure 2 Data acquisition for device comparison.** Operator A and B annotated landmarks directly on a given skull using a digitizer. The error caused by the digitizer and operator measurement error are entangled. We accounted for this compounding error by letting both operators individually scan the given skull and annotate their own resulting reconstructed 3D model. By doing so, the two errors were also combined for the scanner pipeline, thus, the two devices were comparable. For both devices, landmark annotation was repeated. The datasets obtained by operator A using a Microscribe digitizer were collected about one year earlier than all remaining datasets. This left us with eight datasets, each containing 22 landmark configurations. We call this set of eight datasets the *Device Comparison Dataset*.

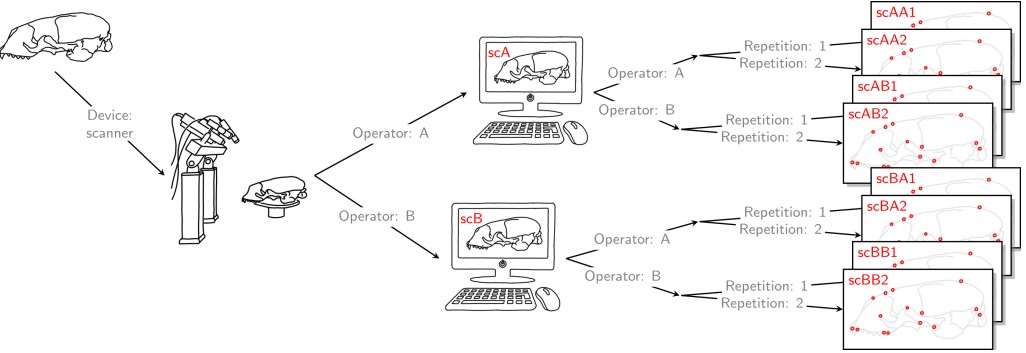

**Figure 3 Extended scanner data acquisition.** Both operators individually scanned a given skull, and they both annotated landmarks on the two resulting reconstructed 3D models. Landmark annotation was repeated. This left us with eight scanner datasets, each containing 22 landmark configurations. We call this set of eight scanner datasets the *Extended Scanner Dataset*. In contrast to the *Device Comparison Dataset* (Fig. 2), we could disentangle the scanning-related error from the operator measurement error.

covered by the stand, and landmarks were only placed from a single perspective. The importance of landmark choice to reduce measurement error is highlighted in *Fruciano et al. (2017)* and *Marcy et al. (2018)*.

In order to align landmark configurations to a common frame of reference, a generalized Procrustes analysis (GPA) (*Gower, 1975*; *Ten Berge, 1977*; *Goodall, 1991*) was performed, in which the configurations were scaled to unit centroid size, followed by projection to tangent

space. GPA eliminates all geometric information (size, position and orientation) that is not related to shape. It uses the minimization of the Procrustes distance as a criterion. In all shape analyses, the Procrustes tangent coordinates were used as shape variables. Whenever we did statistical analyses on subsets of the whole data set, a general Procrustes analysis was only applied to these subsets before doing the statistical analyses. GPA and all statistical analyses were computed in R version 3.5.3 using the packages geomorph (*Adams & Otrola-Castillo, 2013*) and Morpho (*Schlager, 2017*).

In order to investigate measurement error, we computed Procrustes distances between devices, between operators, within operators (i.e., between landmark replica), and between scans. The Procrustes distance is defined as the sum of distances between corresponding landmarks of two GPA-aligned shapes. This allowed us to investigate the extent of differences in the total shape of the same specimen in various ways: measurement by (a) the same operator using a different device (between-device error), (b) different operators using the same device (between-operator error), (c) the same operator using the same device (within-operator error), and (d) the same operator on different scans of the same skull (between-scan error). For (a), (b) and (c), we used the *Device Comparison Dataset*, whereas (d) is based on the *Extended Scanner Dataset*.

To assess the relative amount of measurement error resulting from the different error sources, we ran different Procrustes ANOVAs (*Goodall, 1991*; *Klingenberg & McIntyre, 1998*; *Klingenberg, Barluenga & Meyer, 2002*; *Collyer, Sekora & Adams, 2015*). This approach allowed us to estimate the relative contribution of both biological variation, and variation induced by source-specific measurement error (non-biological) to total shape variation. We also computed repeatability (*Arnqvist & Mårtensson, 1998*; *Fruciano, 2016*) based on Procrustes ANOVA mean squares. Repeatability measures the variability of repeated measurements of the same specimen relative to biological variation. It takes values between zero and one, and the closer to one, the smaller the relative amount of measurement error. In the first step, we analysed one error source at a time by performing Procrustes ANOVAs on pairwise dataset comparisons (similar to *Fox, Veneracion & Blois, 2020*), and computing repeatability for each dataset comparison. Based on 20 unique pairwise dataset comparisons on the *Device Comparison Dataset*, and 8 unique pairwise dataset comparisons on the *Extended Scanner Dataset*, we computed mean repeatability for between-device, between-operator, within-operator, and between-scan comparison. In the second step, we conducted a series of Procrustes ANOVAs cumulatively on multiple datasets. To quantify the measurement error induced by device, and to compare the two devices, we used the *Device Comparison Dataset*. We obviously have a crossed data structure, but in parallel to other authors (e.g., *Fox, Veneracion & Blois, 2020*; *Robinson & Terhune, 2017*), we decided to analyse the data using a nested model. This corresponds to simultaneously testing for main effects and higher order interaction effects. We applied the following nested hierarchical structure: Specimen > Device > Operator > Landmark replica. We also computed repeatabilities for single landmark replicas. To assess the level of scanning-related error, we used the *Extended Scanner Dataset*, and applied the following nested hierarchical structure: Specimen > Scan replica > Operator > Landmark replica.

To investigate whether device and operator introduce general shape bias, i.e., non-random variation in mean shape, we performed a PCA on the residuals computed by subtracting specimen means from the individual landmark configurations in the *Device Comparison Dataset* (c.f. Fig. 2). This approach removes biological variation, thus making it easier to visualize the variation due to measurement error (*Fruciano et al., 2017*; *Fruciano et al., 2019*). We additionally computed the mean shape for both devices and operators in order to visualize differences.

## RESULTS

We found that the distribution of Procrustes distances between devices is comparable to that between operators (Fig. 4). In general, Procrustes distances between devices or between operators exhibit larger values than those within operators or between scans. Moreover, Procrustes distances between scans are distributed similarly to those within operators. The median of the Procrustes distances within operators is similar for both devices, providing evidence that the digitizer and scanner exhibit similar precision. However, there is a longer tail for the scanner-based data. The Procrustes distances between operators using the scanner pipeline seem to be smaller than when using the digitizer. Measurement differences between devices seem to be similarly distributed for both operators. Our analysis further shows that the median of the Procrustes distances between specimens of grey seal is three to six times larger than the medians of Procrustes distances corresponding to different measurement errors. There is a slight overlap between the smallest 25% of Procrustes distances between specimens and Procrustes distances corresponding to different measurement errors, which is almost completely associated with outliers of the latter.

There are several outliers corresponding to substantial measurement error in all comparisons (Fig. 4). An analysis of these outliers revealed that they are mainly caused by measurements of a few landmarks on six skulls, where three specimens alone (96, 42.23, 14) are responsible for 87.5% of the outliers. On these specimens, some of these landmarks were difficult to place both on the 3D models and on the physical skulls (e.g., landmarks 1, 12/26, 13/27, 14/28, 17/31), as was noted during data acquisition. Our analysis further revealed that the observed landmark measurement errors of the outliers are due to both differences between operators and devices, or a combination of them. We provide some outlier examples in Figs. S1 and S2.

We obtained corresponding results in our analysis based on pairwise Procrustes ANOVAs where we assessed the level of measurement error (Table 1). The most substantial error sources are between-device and between-operator, which are comparable in size to each other. On average, measurement differences between devices explain 4.6%, and between operators 4.5%, respectively, of the total shape variation among datasets in pairwise comparisons. Moreover, error between scans is similar to within-operator error, as they explain 2.0%, and 1.4%, respectively, of pairwise total shape variation on average. The repeatabilities computed for the comparisons by error source exhibit a similar pattern.

Running a Nested Procrustes ANOVA on the whole *Device Comparison Dataset*, thereby testing the two devices on equal terms, again, gave a comparable result. We found that
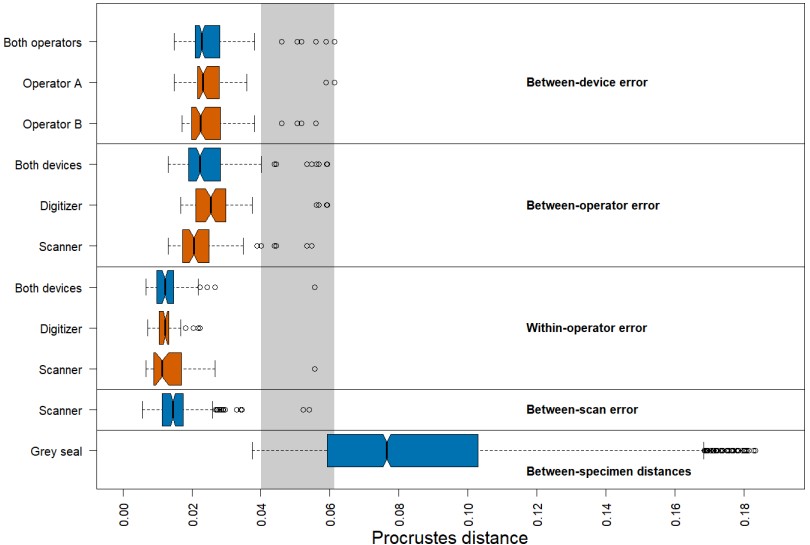

**Figure 4** **Boxplots of Procrustes distances.** Computation of Procrustes distances between devices ($n = 176$); between devices for a given operator ($n = 88$, each); between operators ($n = 176$); between operators for a given device ($n = 88$, each); within operators ($n = 88$); within operators for a given device ($n = 44$, each); between scans ($n = 176$), and between specimens ($n = 3696$). The thick bars represent the median, boxes display the interquartile range, and the whiskers extend to 1.5 times the interquartile range. Outliers are represented by circles. The boxplot colours indicate whether a boxplot is based on all Procrustes distances for a given error source (blue) or on a subset (red). The grey area highlights the range of Procrustes distances at which observed errors are outliers for all error sources. The computation of Procrustes distances between scans is based on the *Extended Scanner Dataset*, whereas all other computations are based on the *Device Comparison Dataset*.

**Table 1** **Pairwise Procrustes ANOVAs on shape.** For a specific error source, Procrustes ANOVA was run on all unique paired datasets, and repeatability was computed. We report mean Procrustes ANOVA residual $R^2$ (Mean Rsq) and mean repeatability (Mean R). For the comparison between scans, the *Extended Scanner Dataset* was used, whereas all other comparisons were based on the *Device Comparison Dataset*. A more detailed table including all pairwise Procrustes ANOVAs can be found in Table S1.

| Error source | Mean Rsq | Mean R |
|---|---|---|
| Between-device | 0.046 | 0.954 |
| Between-operator | 0.045 | 0.955 |
| Within-operator | 0.014 | 0.986 |
| Between-scan | 0.020 | 0.981 |

the device accounts for 2.5% of total shape variation, while between-operator and within-operator error explain 3.7% and 1.4% of the total shape variation (Table 2A). Most of the total shape variation (92.4%) is explained by biological variation among grey seal specimens.

Computing Nested Procrustes ANOVAs for both devices separately, we found that the variation in residuals contributes to a similarly small fraction of total shape variation for both devices (Tables 2B and 2C). As the residuals correspond to differences in landmark replica (i.e., within operators), this indicates that the precision is similar for both the

**Table 2** **Nested Procrustes ANOVA on shape for device comparison.** We applied the following nested hierarchical structure: Specimen > Device > Operator > Landmark replica. (A) All datasets. (B) Only scanner-based datasets. (C) Only digitizer-based datasets. The R-squared values (Rsq) give an estimate of the relative contribution of each factor to the total shape variation. Repeatabilities are for landmark replica only.

| Variables | Df | SS | MS | Rsq | F | Z | Pr(>F) | R |
|---|---|---|---|---|---|---|---|---|
| **A.** *Device Comparison Dataset* | | | | | | | | |
| Specimen | 21 | 0.602 | 0.030 | 0.924 | 272.705 | 21.092 | 0.001 | |
| Specimen:Device | 22 | 0.016 | 0.001 | 0.025 | 6.916 | 26.363 | 0.001 | |
| Specimen:Device:Operator | 44 | 0.025 | 0.001 | 0.037 | 5.260 | 27.812 | 0.001 | |
| Residuals (Landmark replica) | 88 | 0.010 | 0.000 | 0.014 | | | | |
| Total | 175 | 0.671 | | | | | | |
| **B.** *Device Comparison Dataset*: **Scanner** | | | | | | | | |
| Specimen | 21 | 0.316 | 0.015 | 0.954 | 114.059 | 17.562 | 0.001 | 0.983 |
| Specimen:Operator | 22 | 0.009 | 0.000 | 0.028 | 3.230 | 24.287 | 0.001 | |
| Residuals (Landmark replica) | 44 | 0.006 | 0.000 | 0.018 | | | | |
| Total | 87 | 0.332 | | | | | | |
| **C.** *Device Comparison Dataset*: **Digitizer** | | | | | | | | |
| Specimen | 21 | 0.315 | 0.015 | 0.942 | 177.721 | 19.191 | 0.001 | 0.989 |
| Specimen:Operator | 22 | 0.016 | 0.001 | 0.047 | 8.441 | 25.747 | 0.001 | |
| Residuals (Landmark replica) | 44 | 0.004 | 0.000 | 0.011 | | | | |
| Total | 87 | 0.334 | | | | | | |

**Table 3** **Nested Procrustes ANOVA on shape for identifying scanning-related error.** We applied the following nested hierarchical structure: Specimen > Scan replica > Operator > Landmark replica. The R-squared values (Rsq) give an estimate of the relative contribution of each factor to the total shape variation.

| Variables | Df | SS | MS | Rsq | F | Z | Pr(>F) |
|---|---|---|---|---|---|---|---|
| *Extended Scanner Dataset* | | | | | | | |
| Specimen | 21 | 0.628 | 0.030 | 0.951 | 261.095 | 20.324 | 0.001 |
| Specimen:Scan replica | 22 | 0.005 | 0.000 | 0.007 | 1.895 | 23.964 | 0.001 |
| Specimen:Scan replica:Operator | 44 | 0.017 | 0.000 | 0.026 | 3.451 | 27.049 | 0.001 |
| Residuals (Landmark replica) | 88 | 0.010 | 0.000 | 0.015 | | | |
| Total | 175 | 0.660 | | | | | |

digitizer and the scanner pipeline, and high for grey seal skulls. This result is also reflected in the repeatabilities for landmark replica only, which is 0.983 for the scanner-based datasets, and 0.989 for the digitizer-based datasets, respectively. We again find evidence that the variation between operators contributes slightly more to total shape variation among digitizer-based datasets (4.7%) than among scanner-based datasets (2.8%).

Running a Nested Procrustes ANOVA on the *Extended Scanner Dataset*, thereby disentangling the scanning-related error from the operator error, we found that out of all identified error sources, scanning-related error is smallest and explains only 0.7% of among-dataset shape variation (Table 3).

Our analyses on general shape bias, which are based on the residuals computed by subtracting specimen means from the individual landmark configurations in the *Device Comparison Dataset*, visually indicate that both device and operator introduce bias, i.e.,

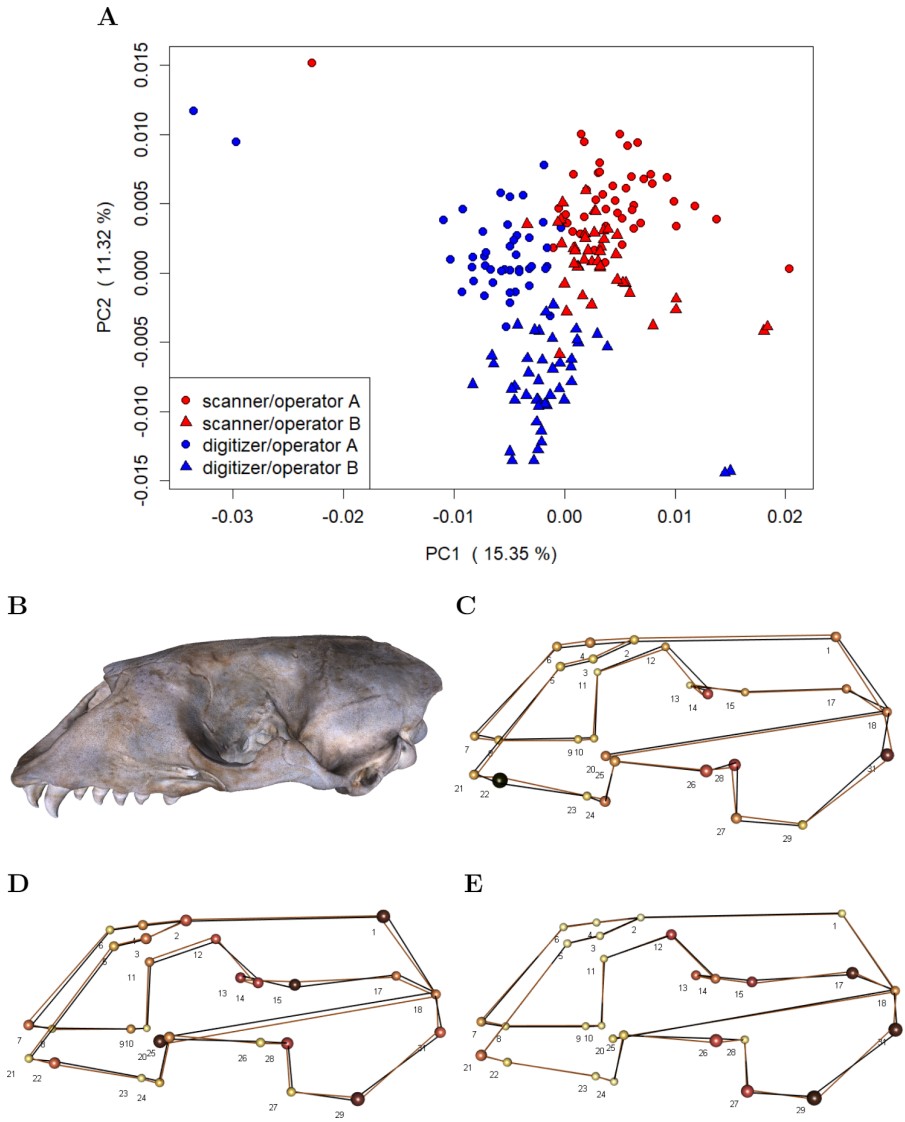

**Figure 5 Pairwise Procrustes ANOVAs on shape.** (A-E) For a specific error source, Procrustes ANOVA was run on all unique paired datasets, and repeatability was computed. We report mean Procrustes ANOVA residual $R^2$ (Mean Rsq) and mean repeatability (Mean R). For the comparison between scans, the *Extended Scanner Dataset* was used, whereas all other comparisons were based on the *Device Comparison Dataset*. A more detailed table including all pairwise Procrustes ANOVAs can be found in Table S1.

non-random error (Fig. 5). The scatterplot of the scores along the first two principal components (Fig. 5A) on the residuals from specimen means shows that the scanner pipeline and digitizer methods can be roughly linearly separated along the first principal components. Moreover, in case of the digitizer method, one can roughly categorize operator A and B along the second principal component, whereas the scores of operator A and B are slightly more overlapping for the scanning pipeline. This is further evidence that the operator-bias is larger on the digitizer than on the scanner. Between devices, we find the largest differences in average shape (Fig. 5C) for landmarks that are less clearly defined,

including e.g., landmarks at the alveolar margins of the teeth (e.g., 22), caudal landmarks (e.g., 17/31, 18), and around the orbits (e.g., 12/26, 14/28). We further observe that for symmetric pairs of landmarks, the between-device error is not reflecting this symmetry: In two-thirds of the cases, the error is larger on the lateral left side. Moreover, we find evidence that on the 3D model, the surface spanned by landmarks 2–6, which are located on the sutures of the nasal, is translated inwards from the analogous surface measured using the digitizer. Between operators (Figs. 5D and 5E), larger shape differences are predominant in similar regions as between devices, however, with less bias at the alveolar (teeth) landmarks. In general, between-operator shape differences again seem to be larger using the digitizer. For the scanner pipeline, the smallest between-operator bias is for landmarks located on the sutures of the nasal (all Type I), which is in contrast to landmark measurement using a digitizer. Similar to between-device error, we note that between-operator error does not seem to be symmetric for some symmetric landmarks (12/26, 13/27 and 17/31) when using the digitizer.

## DISCUSSION

In this study, we investigated whether a 3D structured light scanning setup is a viable alternative to a Microscribe digitizer, one of the most popular choices for fixed landmark measurement of skeletal material within geometric morphometrics, by assessing measurement error involved in data collection on grey seal skulls. We specifically accounted for the interaction between operator and digitizer during data collection by introducing a comparable compounding effect between operator and machine for the scanner pipeline. Without doing this, we might have introduced bias in favour of the SeeMaLab scanner over a digitizer. In our analysis, we compared within- and between-operator error obtained with the two devices. Additionally, we quantified the variation introduced by measurement device and compared it to other sources of variation. Furthermore, we illustrated that both measurement method and operator introduce general shape bias (i.e., non-random variation in mean shape). Finally, we quantified the scanning-related error (i.e., the error introduced by small differences in the scanning pipeline).

Overall, our analysis suggests that the SeeMaLab surface scanner provides a good alternative to the Microscribe digitizer for geometric morphometrics. We found comparably small within-operator errors, which implies that the precision of the SeeMaLab scanner compares with the precision of the digitizer. This was also reflected in the repeatability scores for landmarking replica (scanner: 0.98, digitizer: 0.99), which were similar to previously reported values obtained on human skulls (digitizer: 0.99; *Badawi-Fayad & Cabanis, 2007*), kangaroo-size skulls (overall repeatability for laser scanners and photogrammetry (pooled): 0.78−0.96; *Fruciano et al., 2017*), and bat skulls (laser scanner, μCT scanner and photogrammetry (pooled): 0.97; *Giacomini et al., 2019*). *Marcy et al. (2018)* have found a smaller repeatability of 0.64 for laser scanner measurements of fixed landmarks using small rodent skulls.

From the distribution of Procrustes distances, which were computed between landmark replica measured by the same operator using the same device, we could see that for many

skulls, operators were more consistent in placing landmarks on the 3D digital model than when digitizing the physical skull. However, the long tail of the distribution for the scanner-based data showed that for a few skulls, the operators were less consistent in repeating annotation of some landmarks than when using the Microscribe digitizer. The main drivers of the observed inconsistencies were a few landmarks (symmetric pairs 13/27 and 14/28), which both operators found difficult to digitize. Leaving these landmarks out, the long tail disappeared, and there was evidence that landmark placement using the scanner pipeline was more precise than using the digitizer, which is in line with *Robinson & Terhune (2017)*. As opposed to this, *Sholts et al. (2011)* have found that Type I and II landmarks can be more precisely identified using a digitizer compared to a surface scanner, which might be partly explained by the fact that for some specimens, the authors reported difficulties in locating the sutures on the 3D model due to surface alterations. Our results underline the importance of carefully choosing the landmarks when using a scanner, as previously has been pointed out by *Fruciano et al. (2017)*. Another approach could be to repeat measurements of difficult landmarks on the 3D models, as taking averages of repeated measurements reduces within-operator error (*Arnqvist & Mårtensson, 1998*; *Fruciano, 2016*).

In general, we found that between-operator error is larger than within-operator error, as previously shown by other studies (*Robinson & Terhune, 2017*; *Shearer et al., 2017*; *Wilson, Cardoso & Humphrey, 2011*), and that between-operator error is systematic, which is in line with *Fruciano et al. (2017)*. Our results show that between-operator error is smaller in the scanner-based dataset than in the digitizer-based dataset, which is in contrast to *Robinson & Terhune (2017)*, who found similar error for two such devices. Part of the reason for the difference could be that in this study, we investigated larger skulls, which resulted in relatively higher-resolution scans compared to *Robinson & Terhune (2017)*, who reported a substantial variation in the resolution of their scans. When comparing different scanners, we would expect between-operator error to be more similar, as placing landmarks on 3D models resulting from different scanners is more similar than placing landmarks using a digitizer. *Shearer et al. (2017)* found that between-operator error is independent of scan type, whereas *Fruciano et al. (2017)*, similarly to our study, reported differences. One potential explanation for the observed difference in between-operator error between devices could be that the digitizer might have been used differently by the operators in terms of placing landmarks from slightly different angles. This could have resulted in slightly different offsets as the Microscribe tip is a bit blunt. Furthermore, there might also be differences in the digitizer's joints, which might have been used in slightly different ways by the two operators. In addition, the operators might have investigated the landmarks from a more similar angle using the scanner pipeline, as they could rotate the 3D model and zoom in, whereas an operator had to change the position of her/his body (e.g., bending down) to be able to place a landmark with the digitizer, which could be done in many different ways. In line with this reasoning, we found some evidence that the placement of the digitizer with respect to the skull had an influence on between-operator error, as we could see that between-operator error seemed to be more bilaterally different for some symmetric landmarks (12/26, 13/27 and 17/31) when using a digitizer compared

to the scanner (Article S1, Fig. 1 shows our measurement setup using a digitizer). Another interesting result is that the smallest between-operator bias was obtained for landmarks located on intersecting sutures of the nasal in case of the scanner pipeline, which was not the case for landmark measurement using a digitizer. One additional explanation is that landmark coordinates were bound to the surface of the 3D model, whereas landmark coordinates measured using a digitizer might in practice have been digitized off the skull, providing a further dimension of error for the Microscribe.

We also found systematic error between the two devices. This is in line with *Fruciano et al. (2017)*, who documented the presence of non-random error between different 3D capture modalities. Specifically, we found the largest differences in average shape for landmarks that are less clearly defined (Type II). The placement of these landmarks was to a larger extent driven by an operator's interpretation, such that device-specific characteristics might have led to a larger bias. For example, landmarks 1 and 14/28 are points of maximum curvature located at tips of processes. An operator could locate these landmarks both by visual and tactile means on the physical skull, which was quite helpful, whereas the tactile component is lost when using a 3D digital model. Moreover, it might have been more difficult to place landmarks on a computer screen compared to using a digitizer, as the computer image is two-dimensional. Future improvements, such as the use of Virtual Reality, might make the 3D view of the digital model tangible to a level comparable to reality, while also providing all the advantages of a digital model. On a few skulls, landmarks 14/28 were located on a very pointy tip. There is no guarantee that such a thin structure is scanned properly. Interestingly, for two-thirds of the symmetric landmark pairs, the between-device error seemed to be larger for landmarks located on the left side. Similarly to the analysis of between-operator error, this asymmetry might have been caused by the placement of the digitizer relative to the skull, which was on the dorsal side. One example is landmark 22, which is located at the posterior alveolar margin of canine. However, we cannot exclude the possibility that the 3D models were biased around the teeth due to the observed problems with reflection and being out of focus during scanning. Another source of error may be exemplified by landmarks 17/31 and 18, which are caudal landmarks. In our digitizer setup, these landmarks were placed in an awkward body position where the operator was bending down, and pressing the foot pedal at the same time. This is in sharp contrast to placing these landmarks on the 3D digital model, where the operator could focus on the task in a relaxed body position as manipulation of the 3D model happened on the computer screen. We further observed that the surface spanned by the Type I landmarks 2-6 on the 3D model was translated inwards compared to the surface spanned by these landmarks when measured using the digitizer. This might be explained by the fact that these landmarks are located at intersections of bones at sutures. On the 3D model, these landmarks were automatically placed on the surface, whereas the digitizer tip did not necessarily have to touch the surface of the skull. Another explanation could be that the 3D model differed in size from the physical skull. However, when comparing various inter-landmark distances from Microscribe and scanner, we did not find any evidence for a difference in skull size. In general, the colour model of the 3D digital models is imperfect, which might also have

led to differences in landmark placement between the two devices. However, the colour model could easily be improved.

Our Procrustes ANOVA results show that the factors *measurement device*, *operator* and *landmark replica* all are statistically significant and of similar importance, but are still rather minor sources of total shape variation, compared to the biological variation of the grey seal skulls. Our analysis of pairwise Procrustes distances revealed a similar pattern. Other studies on similarly sized skulls have found a comparably large biological variation (e.g., on human and great ape skulls, *Martnez-Abadas et al., 2012*; *Singh et al., 2012*). In contrast to our study, *Robinson & Terhune (2017)* found evidence that patterns among specimens at the intraspecific level may be obscured by scanner-Microscribe differences and/or between-operator error. However, their result is based on the much smaller species *Callicebus* (male body mass = 1.02 kg), than the grey seal (male body mass = 170–310 kg). As it generally can be assumed that the effect and relative contribution of measurement error becomes larger as true biological variation becomes lower (*Fruciano, 2016*; *Fruciano et al., 2017*; *Fruciano et al., 2019*), we repeated the same analysis based on only the 17 specimens from the Baltic Sea population, excluding the five specimen from the western North Atlantic population, however, we obtained a similarly large biological variation (Table S2).

In contrast to previous studies comparing 3D surface scanners to other devices in the context of geometric morphometrics data collection (e.g. *Sholts et al., 2011*; *Algee-Hewitt & Wheat, 2016*; *Fruciano et al., 2017*; *Robinson & Terhune, 2017*; *Shearer et al., 2017*; *Marcy et al., 2018*; *Giacomini et al., 2019*), we used an open-source system (*Eiriksson et al., 2016*) rather than a commercial scanner. Using such an open-source system has the advantage of having full control over the scanning pipeline. In contrast to using commercial scanners, we could freely decide how to clean the noisy raw scans, and which reconstruction algorithm we wanted to use, thus allowing us to minimize scanning-related measurement error. As this system comes with an increased amount of freedom, an operator had to make some choices with respect to scanning parameters, such as exposure time, or the distance between the scanner and the specimen to scan, even when following a general scanning protocol. Our analysis of the *Extended Scanner Dataset* revealed that the error due to these scanning-related differences was significant, but smaller than other errors, also within-operator error, which is linked to precision. Our analysis further suggests that 3D digital models, which are based on scans from different operators that followed a general protocol, were only a minor source of total shape variation. We note that even though the distance between scanner and the specimen is linked to measurement error due to optical distortion (an example of presentation error (*Fruciano, 2016*)), we could not systematically analyse this error source, as parameter choice was at the discretion of the operators.

We note that we could have constructed the *Device Comparison Dataset* differently by using the datasets where each operator placed the landmarks on the 3D models reconstructed from the scans made by the other operator (datasets scAB1, scAB2, scBA1, scBA2) instead of their own scans (datasets scAA1, scAA2, scBB1, scBB2). By doing so, we noticed a trend of slightly lower repeatabilities when comparing devices by means of

pairwise Procrustes ANOVA (Table S3), however, we did not find significant differences in the Nested Procrustes ANOVA analyses (Table S4).

## CONCLUSIONS AND RECOMMENDATIONS

In this study, we used a carefully chosen setup, which allowed the investigation of multiple sources of measurement error (within- and between-operator, between-device, and between-scan), and we accounted for the compounding of error between operator and digitizer by having each operator digitizing separate scans.

Our results suggest that the SeeMaLab scanner pipeline is well suited for landmark measurement on grey seal skulls, and that it provides a viable alternative to a Microscribe digitizer for geometric morphometrics at intraspecific level for skulls of similar sizes. On average, we obtained a similar precision with both devices. However, we found evidence that for a few skulls, operators were less consistent in replicating annotation of landmarks on the scanner-based 3D models compared to the digitizer. This was linked to a few difficult landmarks. By omitting difficult landmarks, we found that the scanner pipeline resulted in a higher precision compared to the digitizer, and the problem that an operator was not always consistently placing landmarks was overcome. Thus, we would recommend any user of a scanner pipeline for geometric morphometrics to do a pre-study to identify difficult to place landmarks, and develop a strategy of how to deal with these landmarks (e.g., leave them out, or repeatedly measure them and use an average) in order to obtain a precision as high as possible. We would like to stress that both a 3D surface scanner and a digitizer can be used for geometric morphometrics, and that they both come with their respective strengths and limitations.

Our study further points out that between-operator error, which is one of the largest measurement error sources, is biased and seems to be smaller when using the scanner pipeline than the digitizer. Especially for landmarks placed on intersecting sutures, there is evidence that between-operator error is smaller using the scanner. Moreover, we found systematic differences between devices, mainly driven by landmarks that are not clearly defined. For our data-set of relatively large skulls, where operator and device are not contributing much to total shape variation, it might not be a problem combining data from two operators, or indeed two devices. However, since we observed systematic error between both devices and operators, we would not advise to combine digitizer and scanner data, or data from different operators when the expected biological variation is small without conducting a pre-study and testing the landmarks for repeatability across platforms and operators. For any study where it will be necessary to combine landmark measurements from different operators, the scanner pipeline will be better suited due to the smaller between-operator error.

In contrast to previous studies, which compared the use of 3D surface scanners to other 3D capture modalities or digitizers, we used the SeeMaLab scanner, which is an open-source system developed at the Technical University of Denmark. The use of such a system has the advantage of full control over the process of generating 3D models. The scanning-related error, i.e., the error due to different scanning parameter choices, is very small compared to
all other error sources for our data set. Thus, we do not expect the landmark measurements to vary substantially between different scans. Furthermore, in principle, different operators can scan different specimens without introducing significant error, as long as they all follow the same scanning protocol. This, again, underlines the importance of a pre-study to set up a common scanning protocol in order to obtain satisfying 3D models of the specimens of interest. It would be interesting to systematically investigate presentation error due to optical distortion or different reconstruction algorithms and compare it to other error sources in future work. Since the SeeMaLab scanner so far mainly has been used as a research platform, we note that the scanner comes with some practical issues (e.g., profound knowledge required; manual steps; reflection and focus issues at specimen teeth), however, the scanner pipeline could easily be improved by automating parts of the pipeline and implementing quality control and high-dynamic range techniques.

## ACKNOWLEDGEMENTS

The authors would like to thank Daniel Klingberg Johansson from the Natural History Museum of Denmark, and Mia Valtonen from the Institute of Biotechnology at University of Helsinki for giving us access to the grey seal collections. We would further like to thank Florian Gawrilowicz for his assistance with the 3D model creation part, and for providing us with his code for non-rigid point cloud registration. In addition, the authors would like to thank M. Pilar J. Stella for her contribution to the pre-study which goal was to set up a scanning protocol in order to obtain satisfying 3D models of grey seal skulls.

## APPENDIX

**Table A1** List of selected specimens (*Halichoerus grypus*).

| Institution | Specimen | Population | Sex | Age (y) |
|---|---|---|---|---|
| NHMD | 14 | Baltic Sea | NA | NA |
| NHMD | 15.7 | Baltic Sea | NA | NA |
| NHMD | 42.11 | Baltic Sea | NA | NA |
| NHMD | 42.23 | Baltic Sea | NA | NA |
| NHMD | 69.1 | Baltic Sea | NA | NA |
| NHMD | 71.2 | Baltic Sea | NA | NA |
| NHMD | 95 | Baltic Sea | NA | NA |
| NHMD | 96 | Baltic Sea | male | NA |
| NHMD | 101.11 | Baltic Sea | NA | NA |
| NHMD | 134 | Baltic Sea | NA | NA |
| NHMD | 185 | Baltic Sea | NA | NA |
| NHMD | 223 | Baltic Sea | NA | NA |
| NHMD | 323 | Baltic Sea | NA | NA |
**Table A1** (*continued*)

| Institution | Specimen | Population | Sex | Age (y) |
|---|---|---|---|---|
| NHMD | 417.2 | Baltic Sea | NA | NA |
| NHMD | 417.3 | Baltic Sea | NA | NA |
| NHMD | 417.4 | Baltic Sea | NA | NA |
| NHMD | 664 | Baltic Sea | NA | NA |
| UH | C7-98 | Western North Atlantic | male | 35 |
| UH | C10-98 | Western North Atlantic | female | 22 |
| UH | C13-98 | Western North Atlantic | female | 14 |
| UH | C16-98 | Western North Atlantic | female | 17 |
| UH | C17-98 | Western North Atlantic | female | 18 |

**Notes.**

NHMD, Natural History Museum of Denmark; UH, University of Helsinki, Finland; NA, not available.
Preference was given to larger skulls, assuming they were adults.

**Table A2  List of anatomical landmarks.** We used 31 fixed anatomical landmarks (L = left, R = right). 6 landmarks are of Type I, and 25 of Type II.

| Landmark description | Name | Type |
|---|---|---|
| Apex of supraoccipital | 1 | II |
| Caudal apex of nasal | 2 | I |
| Intersection of maxillofrontal sutureand nasal (L, R) | 3, 4 | I |
| Intersection of maxillapremaxilla suture and nasal (L, R) | 5, 6 | I |
| Anterior point of canine (L, R) | 7, 21 | II |
| Posterior point of canine (L, R) | 8, 22 | II |
| Anterior point of last (fifth) molar (L, R) | 9, 23 | II |
| Posterior point of last molar (L, R) | 10, 24 | II |
| Anterior apex of jugal (L, R) | 11, 25 | II |
| Dorsal apex of jugaltemporal suture (L, R) | 12, 26 | II |
| Posterior apex of jugal (L, R) | 13, 27 | II |
| Ventral apex of orbital socket (L, R) | 14, 28 | II |
| Apex of auditory process (L, R) | 15, 29 | II |
| Posteriordorsal apex of suture above lateral apex of condyle (L, R) | 16, 30 | I |
| Right lateral apex of condyle (L, R) | 17, 31 | II |
| Dorsal apex of foramen magnum | 18 | II |
| Ventral apex of foramen magnum | 19 | II |
| Ventral point of intersection between palatines and maxillas | 20 | I |

**Table A3  List of 3D models of grey seal (*Halichoerus grypus*) skulls used in this study and their source.** Both operator A and B were scanning grey seal skulls collected at the Natural History Museum of Denmark (NHMD) and University of Helsinki (UH) using the SeeMaLab scanner.

| Institution | Specimen | Scan operator | Source (MorphoSource identifiers) |
|---|---|---|---|
| NHMD | 14 | A | https://doi.org/10.17602/M2/M357280 |
| NHMD | 14 | B | https://doi.org/10.17602/M2/M356856 |

| Institution | Specimen | Scan operator | Source (MorphoSource identifiers) |
|---|---|---|---|
| NHMD | 15.7 | A | https://doi.org/10.17602/M2/M357311 |
| NHMD | 15.7 | B | https://doi.org/10.17602/M2/M357639 |
| NHMD | 42.11 | A | https://doi.org/10.17602/M2/M357658 |
| NHMD | 42.11 | B | https://doi.org/10.17602/M2/M357654 |
| NHMD | 42.23 | A | https://doi.org/10.17602/M2/M357667 |
| NHMD | 42.23 | B | https://doi.org/10.17602/M2/M357804 |
| NHMD | 69.1 | A | https://doi.org/10.17602/M2/M364239 |
| NHMD | 69.1 | B | https://doi.org/10.17602/M2/M357809 |
| NHMD | 71.2 | A | https://doi.org/10.17602/M2/M364243 |
| NHMD | 71.2 | B | https://doi.org/10.17602/M2/M357814 |
| NHMD | 95 | A | https://doi.org/10.17602/M2/M364271 |
| NHMD | 95 | B | https://doi.org/10.17602/M2/M357819 |
| NHMD | 96 | A | https://doi.org/10.17602/M2/M364247 |
| NHMD | 96 | B | https://doi.org/10.17602/M2/M357830 |
| NHMD | 101.11 | A | https://doi.org/10.17602/M2/M364251 |
| NHMD | 101.11 | B | https://doi.org/10.17602/M2/M357885 |
| NHMD | 134 | A | https://doi.org/10.17602/M2/M364255 |
| NHMD | 134 | B | https://doi.org/10.17602/M2/M357890 |
| NHMD | 185 | A | https://doi.org/10.17602/M2/M364259 |
| NHMD | 185 | B | https://doi.org/10.17602/M2/M357895 |
| NHMD | 223 | A | https://doi.org/10.17602/M2/M364279 |
| NHMD | 223 | B | https://doi.org/10.17602/M2/M357908 |
| NHMD | 323 | A | https://doi.org/10.17602/M2/M364263 |
| NHMD | 323 | B | https://doi.org/10.17602/M2/M357909 |
| NHMD | 417.2 | A | https://doi.org/10.17602/M2/M364267 |
| NHMD | 417.2 | B | https://doi.org/10.17602/M2/M357915 |
| NHMD | 417.3 | A | https://doi.org/10.17602/M2/M364275 |
| NHMD | 417.3 | B | https://doi.org/10.17602/M2/M357923 |
| NHMD | 417.4 | A | https://doi.org/10.17602/M2/M364283 |
| NHMD | 417.4 | B | https://doi.org/10.17602/M2/M357925 |
| NHMD | 664 | A | https://doi.org/10.17602/M2/M364287 |
| NHMD | 664 | B | https://doi.org/10.17602/M2/M357939 |
| UH | C7-98 | A | https://doi.org/10.17602/M2/M364293 |
| UH | C7-98 | B | https://doi.org/10.17602/M2/M357940 |
| UH | C10-98 | A | https://doi.org/10.17602/M2/M364299 |
| UH | C10-98 | B | https://doi.org/10.17602/M2/M357952 |
| UH | C13-98 | A | https://doi.org/10.17602/M2/M364294 |
| UH | C13-98 | B | https://doi.org/10.17602/M2/M357958 |
| UH | C16-98 | A | https://doi.org/10.17602/M2/M364307 |
| UH | C16-98 | B | https://doi.org/10.17602/M2/M357963 |
| UH | C17-98 | A | https://doi.org/10.17602/M2/M364302 |
| UH | C17-98 | B | https://doi.org/10.17602/M2/M357971 |

### Funding

The authors received no funding for this work.

### Competing Interests

The authors declare there are no competing interests.

### Author Contributions

- Dolores Messer conceived and designed the experiments, performed the experiments, analyzed the data, prepared figures and/or tables, authored or reviewed drafts of the paper, and approved the final draft.
- Michelle S. Svendsen conceived and designed the experiments, performed the experiments, authored or reviewed drafts of the paper, and approved the final draft.
- Anders Galatius and Knut Conradsen conceived and designed the experiments, analyzed the data, authored or reviewed drafts of the paper, and approved the final draft.
- Morten T. Olsen, Vedrana A. Dahl and Anders B. Dahl conceived and designed the experiments, authored or reviewed drafts of the paper, and approved the final draft.

### Data Availability

The landmark data is available online at https://doi.org/10.11583/DTU.14178707. The reconstructed 3D models are available at https://www.morphosource.org/projects/000355763. The DOIs of the 3D models used in this study are listed in Table A3.

### Supplemental Information

Supplemental information for this article can be found online at http://dx.doi.org/10.7717/peerj.11804#supplemental-information.

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
