# Peer review of "Measurement error using a SeeMaLab structured light 3D scanner against a Microscribe 3D digitizer"

_PeerJ, doi:10.7717/peerj.11804_

## Round 0.1 · original submission · Major Revisions

Two consistent and very detailed reviews have been received. Please provide point-to-point responses to the reviewers. The revised paper will need to be sent to reviewers for a further review.

Reviewer 1 ·

Basic reporting

1) clear and unambiguous, professional English:
- Overall well written with just some minor words/phrases where clarification would improve the reading.
- lines 24/25: Please rephrase to stress that each operator annotated the landmarks twice each on the skulls and the 3D models.
- lines 64/65: Please rephrase to make the point clearer and not to give the impression that Bookstein compared precisions. Maybe split the sentence in the 3 types of landmarks with the Bookstein reference and then go into the testing of precision.
- line 183: You refer to Figure 1. Do you mean Figure 2? From my point of view referring to the landmark set is confusing at this point.
- line 251: Maybe change …(Fig. 2)… into …(c.f. Fig. 2) to make clearer that not the PCA is shown in Fig. 2 but the setup of the data set.
- lines 304/311: Please consider addressing the landmarks like number 22 differently. Addressing them as teeth landmarks gives the impression that the landmarks are actually located on the teeth and not on the alveolar process of the maxilla as they are shown in Fig. 1.
- lines 318: The wording “most popular choice” is maybe not a as clear as you intend it to be in this context. It seems really to depend on the (sub-) field if the Microscribe is still used today or already abandoned. If you want to focus on your specific field please stress it in this sentence or phrase it more openly to fit a wider audience.

2) Literature references, sufficient field background/context provided:
- lines 48-49: In the first sentence of the introduction you name a few studies. However, none of the old publications about the main advances in the field is mentioned and some references to underline the importance and the versatile use of it as toolkit might be useful as your manuscript can be of interest also in other fields where geometric morphometrics is used (e.g. Bookstein, Rholf, Gunz, Slice, Zelditsh).
- lines 92f.: This very much depends on the used scanner (medical, industrial, synchroton, neutron). For materials in which the radiation exposure is not of importance also rather big structures can be scanned in very high resolution (a few microns). Please rephrase and/or add references.
- line 105: The reconstruction of teeth can also be tricky in cases of surface scanning. You even mention potential reflections in the dental area further down in the manuscript. Based on experience its usually the very smooth and sometimes polished dental surfaces that create the issues. Please consider rewriting to show that this disadvantage is not only limited to photogrammetry.

3) professional article structure, figures, tables. Raw data shared:
- figure 4: Please clarify in the legend what the colors stand for and chose color-blind friendly colors (easy to check for compatibility for example in Photoshop).

- table S1: Would it be possible to provide sex and clarify that all individuals are adult? I assume this information is available and kind of relevant in getting an understanding in how homogenous the shape (cf. sexual dimorphism, ontogeny) is for landmark collection.

- table S3: If you provide a more detailed table in the supplements than please really include all the information like in Table 2.

4) Self-contained with relevant results to hypotheses:
- fulfilled

Experimental design

1) original primary research within Aims and Scope of the journal:
- fulfilled

2) research question well defined, relevant & meaningful. It is stated how research fills an identified knowledge gap.
- fulfilled

3) Rigorous investigation performed to a high technical & ethical standard:
- Overall yes, only a few points that need consideration or maybe just clarification.
- lines 31f; 115f.; 205ff.; 374; 383; 399ff.; 460ff.; : Landmarks used and omitting landmarks
-- I am not sure I get your point of highlighting the omission of landmarks in the abstract. To my understanding you either exclude those landmarks very early in your study (or even pre-study) and mention it in the methods or when testing sources of error make a point of seeing how much error such problematic landmarks may introduce.
-- In addition, in this case you have to balance on how to phrase it very carefully because you used published landmark sets. In other words, by not being able to measure those landmarks, you indirectly question the reproducibility and reliability of the results in the published papers. Therefore, please consider something like “non-reproducible here” or “non-reproducible in this context” in line 205.
-- Further, please clarify why you could not measure landmark 19. Very hidden in the discussion (line 448) you state that some of the measurements were collected with a temporal gap and thereby presumably before the beginning of the here described study. Is this the reason why you stuck to that measurement protocol for the Microscribe? If yes please mention this in the method section (see also my other comment about this).
-- Nevertheless, it would be advantageous to explain in one sentence why you did not make use of taking landmarks from two different perspectives with the Microscribe. It is rather common to take landmarks in at least 2 views and to combine them mathematically into one coordinate system based on reference points (e.g. via the DVLR software). This concerns especially landmark 19, 17/31 and 18
- lines 190 & 448: number of repetitions per operator
-- Please describe when the data was collected in the method section. Were the measurements collected close in time or was there a deliberate break between the two measurement rounds? If you measure only twice super close in time to each other the results might be more similar to each other than with temporal gaps.
-- Furthermore, please provide a reason why each operator measured twice. Multiple measurements (3-5 rounds) might give more robust results in terms of inter- vs. intra- operator errors.

Validity of the findings

1) Impact and novelty not assessed. Negative/inconclusive results accepted. Meaningful replication encouraged where rationale & benefit to literature is clearly stated:
- Yes it is meaningful and clearly states where it adds to still existing gaps in the published studies.

2) All underlying data have been provided; they are robust, statistically sound, & controlled:
- All data necessary has been provided. Nevertheless, there is one aspect that could be added to strengthen the manuscript.
- lines 256ff.; 282ff. & Figure 4: You really nicely discuss the results in terms of pairwise Procrustes distances and how much of the observed variation can be attributed to the different kinds of error. In addition to this, it would have been nice to see the overall distribution of pairwise Procrustes distances in regard to error measurements vs. interindividual. This could be visualized for example as histogram or box plots. What I am aiming at is to see whether the greatest error in all tested categories is smaller than the smallest pairwise Procrustes distance between individuals (here the different seals measured) in the sample. I think it would strengthen your manuscript as it would underline that yes there are sources of error but if being aware of them and controlling them as good as possible that the obtained results will still be meaningful.

3) Conclusions are well stated, linked to original research question & limited to supporting results:
- Over all yes, only one conclusion needs some clarification.
- lines 41f.; 475ff.: combination of data from different operators
-- Of course, combining data from different sources (i.e. different observers/operators) needs some internal checking for compatibility. Nevertheless, results should have the aim to be reproducible that by itself means data collection by a different operator. If the variation of interest is that small that it is overwritten by the interobserver error choosing an entirely different approach or method for addressing these research questions should be considered instead of obscuring the issue by collecting data by single operator.

4) Speculation is welcome, but should be identified as such:
- fulfilled

Additional comments

Dear authors,

I am happy that I was asked to review your manuscript. In the majority of papers intra- and inter-observer as well as inter-method error tests are just carried out as small part hidden in the supplements or reported in a few sentences without presenting the actual data. Therefore, I think it is important to systematically evaluate potential sources of error.

Although it is not within the scope of this paper, but out of curiosity, I wondered if there has been a test of between-device error between the SeeMaLab and other surface scanners?

I hope that I could express my review in a constructive way that allows you to directly identify the critical points and address them.

Reviewer 2 ·

Basic reporting

1. Authors carried out the detailed study on 'Measurement error using a SeeMaLab structured light 3D scanner against a Microscribe 3D digitizer'.
2. The English used is quite clear and technically correct. However, the proof reading of the manuscript need to be done while revising the paper.
3. The authors need to read and cite some more relevant and latest papers to improve the content of the paper.

Experimental design

1. The Introduction section need to be improved further.
2. The research question in this present study needs to be defined in a better possible way.
3. The methods used to carry out the present study need to be more clearly stated.

Validity of the findings

1. The discussion section must be modified and improved.
2. Include major finding of study in conclusion section.
3. The conclusion section should be precise and can be written preferably in bullet points.

Additional comments

1. Authors carried out the detailed study on 'Measurement error using a SeeMaLab structured light 3D scanner against a Microscribe 3D digitizer'.
2. The English used is quite clear and technically correct. However, the proof reading of the manuscript need to be done while revising the paper.
3. The authors need to read and cite some more relevant and latest papers to improve the content of the paper.
4. The Introduction section need to be improved further.
5. The research question in this present study needs to be defined in a better possible way.
6. The methods used to carry out the present study need to be more clearly stated.
7. The discussion section must be modified and improved.
8. Include major finding of study in conclusion section.
9. The conclusion section should be precise and can be written preferably in bullet points.

---

## Round 0.2 · accepted · Accept

The paper can be accepted now.

Reviewer 2 ·

Basic reporting

The revised article looks far better than the original submission.

Experimental design

No comment

Validity of the findings

Good enough. The article may be accepted for suitable publication in its present form.

Additional comments

The revised article looks far better than the original submission. All the comments are addressed properly. The article may be accepted for suitable publication in its present form.